Prepared for submission to JHEP

UTWI-07-2023

# Spacetime Subsystem Symmetries

**Saba Asif Baig, Jacques Distler, Andreas Karch, Amir Raz, Hao-Yu Sun**

*University of Texas, Austin, Physics Department, Austin TX 78712, USA*

*E-mail:* sbaig@utexas.edu, distler@golem.ph.utexas.edu, karcha@utexas.edu,araz@utexas.edu, hkdavidsun@utexas.edu

ABSTRACT: One characteristic feature of many fractonic lattice models, and a defining property of the exotic field theories developed to describe them, are subsystem symmetries including a conservation of not just net electric charge but also electric dipole moments or charges living on submanifolds. So far all such theories were based on internal subsystem symmetries. In this work we generalize the notion of subsystem symmetries to system with subsystem spacetime symmetries with locally conserved energies.

## 1 Introduction

The prime example for a continuum quantum field theory with fractonic "subsystem" symmetry is laid out in the recent work by Seiberg and Shao [1] based on the earlier [2]. Their theory of a 2+1 dimensional real scalar has a large set of subsystem symmetries

$$\phi(t, x, y) \rightarrow \phi(t, x, y) + c_x(x) + c_y(y) \tag{1.1}$$

Since $c_x(x)$ is an arbitrary function of $x$ they have conserved charges $Q_x(x)$ which are independently conserved for all $x$, and similarly for $Q_y(y)$. The only constraint is that the sum of all $Q_x$ is equal to the sum of all $Q_y$ – the constant $c$ is shared between $c_x$ and $c_y$ and so there is only one position independent charge, not two. An action invariant under this full symmetry is easily constructed. Upon deformation, this symmetry can be broken to $c_x$ and $c_y$ being linear functions of $x$ and $y$ only. Instead of an infinite number of charges we are left with only an overall conserved charge (from constant $c$) and an $x$ and $y$ dipole charge (from the linear functions $c_x$ and $c_y$).

All these symmetries are "internal" in the standard sense in that they only act on the fields, not the spacetime coordinates. The dipole here is an electric dipole. One obvious generalization is to look for spacetime subsystem symmetries. In this case one would be looking at locally conserved energy and momentum, and upon deformation conserved gravitational multipole charges instead of electric multipole charges. In a non-relativistic setting time is singled out, so the simplest spacetime fracton symmetry one could be looking for is a fractonic time translation symmetry

$$t \rightarrow t' = t + c(x, y). \tag{1.2}$$

Intriguingly enough, this is symmetry is a subset of the full symmetries preserved by the topological theory of [3] where they allowed and arbitrary function $c(t, x, y)$ of both coordinates as well as time, together with holomorphic reparameterizations of the spatial coordinates. Only requiring the fractonic time translation symmetry (1.2) should allow for more non-trivial dynamics than the topological theories of [3], but still have a more rigid structure than a "standard" non-relativistic theory[1] which only allows constant shifts in $t$. This symmetry also contains the Carrollian contraction of the Poincaré group, which is the $c \to 0$ limit [7–9]. [2]

In this work, we present a very simple continuum field theory exhibiting such a "spacetime subsystem" symmetry and work out some of its properties. We present the Lagrangian in section 2 and analyze its conservation law and transport properties. In section 3 we discuss some potential generalizations and conclude with open questions in section 4.

## 2 Field theories with exotic spacetime symmetries

### 2.1 Action

A very simple Lagrangian with the symmetry (1.2) can be written down for two real scalars $\phi_1$ and $\phi_2$ in $d+1$ dimensions, with generalization to any larger number of real scalars straightforward: [3]

$$\mathcal{L} = \tfrac{1}{2}\dot{\phi}_1^2 + \tfrac{1}{2}\dot{\phi}_2^2 + \tfrac{1}{2}(\dot{\phi}_1 \partial_i \phi_2 - \dot{\phi}_2 \partial_i \phi_1)^2 - V(\phi_1, \phi_2). \tag{2.1}$$

The potential $V$ is arbitrary. To see that the corresponding action is invariant, note that under the transformation (1.2) the spacetime derivatives transform as

$$\begin{aligned} \partial_t &\to \partial_t, \\ \partial_i &\to \partial_i + (\partial_i c)\partial_t. \end{aligned} \tag{2.2}$$

With this, the kinetic terms involving only time derivatives are manifestly invariant, whereas the particular combination of terms in the mixed time and space derivatives containing gradient term was chosen to cancel the terms containing derivatives of $c$. The model has an additional $U(1)$ symmetry rotating $\phi_1$ and $\phi_2$ provided the potential is also symmetric (i.e. $V(\phi_1, \phi_2) = V(\phi_1^2 + \phi_2^2)$.)

### 2.2 Nœther Charges

The main consequence of symmetries are conserved charges. We can work out the standard Nœther currents for time independent space and time translations, the standard momentum and energy density and currents. They turn out to be [4]

---

[1]Incidentally, this is also different from the non-relativistic diffeomorphisms of Newton-Cartan symmetry [4] where we allow for time dependent diffeomorphisms of $x$ and $y$ instead of allowing $x$ and $y$ dependent shifts in $t$. It is also different than investigating what spacetime symmetries fractonic theories have, as was done in [5, 6].

[2]We would like to thank the authors of [10] for bringing this point to our attention.

[3]A previous version of this manuscript contained a sign error in this formula which was pointed out in [11].

[4]A previous version of this manuscript omitted a term in $T_i^j$ which was pointed out in [11].

$$T_0^0 = \mathcal{H}$$
$$T_0^i = 0$$
$$T_i^0 = \pi_1 \partial_i \phi_1 + \pi_2 \partial_i \phi_2 \tag{2.3}$$
$$= \dot{\phi}_1 \partial_i \phi_1 + \dot{\phi}_2 \partial_i \phi_2 + \chi_j \left( \partial_i \phi_1 \partial_j \phi_2 - \partial_j \phi_1 \partial_i \phi_2 \right)$$
$$T_i^j = \chi_i \chi^j - \delta_i^j \mathcal{L}$$

where the Hamiltonian density $\mathcal{H}$ is given by

$$\mathcal{H} = \frac{1}{2} \dot{\phi}_1^2 + \frac{1}{2} \dot{\phi}_2^2 + \frac{1}{2} \chi_i^2 + V(\phi_1, \phi_2) \tag{2.4}$$

with[5]

$$\chi_i = \dot{\phi}_1 \partial_i \phi_2 - \dot{\phi}_2 \partial_i \phi_1. \tag{2.5}$$

The form of the Hamiltonian density displayed here is deceptively simple, as we still express it in terms of the time derivatives of $\phi_i$. When spelled out in terms of the conjugate momentum variables $\pi_i = \delta \mathcal{L}/\delta \dot{\phi}_i$ the Hamiltonian density appears highly non-trivial.

The most interesting aspect of the currents in (2.3) is that the energy current $T_0^i$ vanishes identically. This is, in fact, guaranteed by symmetry. Time translation invariance implies

$$\partial_\mu \left( (\delta t) T_0^\mu \right) = 0. \tag{2.6}$$

This gives the standard current conservation for position independent $\delta t$, but for $\delta_t$ as an arbitrary function of $x$ and $y$, $T_0^i$ has to vanish identically and

$$\partial_t \mathcal{H} = 0. \tag{2.7}$$

As expected, the energy density is locally conserved. Despite the locally conserved energy density, the model has non-trivial dynamics and in the quantum system this implies that the Hilbert space does not locally factorize.

## 2.3 Transport

If energy is locally conserved, is there any dynamics left? Note that while the energy current $T_0^i$ vanishes identically, neither the momentum density $T_i^0$ nor the momentum flux $T_i^j$ do. Their dynamics is still given by the conservation law

$$\partial_0 T_i^0 + \partial_j T_i^j = 0. \tag{2.8}$$

For example, in the hydrodynamic regime we would want to write a constitutive relation for $T_i^j$ in terms of $T_i^0$ in a derivative expansion. This will take the standard Navier-Stokes form,

---

[5]As an aside, it should be noted that the dynamics of the system becomes significantly more transparent if one introduces $\chi_i$ as an independent Hubbard-Stratonovich field, whose algebraic equation of motion yields (2.5). For the simple calculations presented in this work, this is not required.

with derivatives of the conserved energy density $T_0^0$ appearing similar to an external potential. The main difference here is that $T_0^0$ is given by the initial conditions rather than by external forces: at time $t = 0$ one lays down an energy profile which will not evolve in time. This energy profile provides a potential in which momentum flows more or less conventionally.

## 3   Generalizations

As a proof of principle, we constructed a simple model with spacetime sub-system symmetries. While energy is locally conserved, the model still allows for non-trivial evolution in space and time. Our basic construction can be generalized in many interesting directions, let us explicitly demonstrate two:

### 3.1   Reduced Symmetry

The model we constructed so far has the large symmetry of shifts of $t$ by an arbitrary function $c(x, y)$ of $x$ and $y$, corresponding to an energy density that is locally conserved at every point in space. One can wonder whether it is possible to systematically break this symmetry to smaller exotic symmetries more in line with what was done for internal symmetries. For example, one could try to reduce the symmetry to shifts of the form $c_x(x) + c_y(y)$ (with energy conserved along lines) or maybe even $c_x x + c_y y$. An action which formally achieves the former is given by

$$\mathcal{L} = \frac{1}{2}(\partial_t \phi)^2 \pm \frac{g}{3}\phi \left[(\partial_t^2 \phi)(\partial_x \partial_y \phi) - (\partial_x \partial_t \phi)(\partial_y \partial_t \phi)\right] - V(\phi). \tag{3.1}$$

To see this is invariant note that

$$\partial_x \partial_y \phi \to \partial_x \partial_y \phi + (\partial_x c)(\partial_y \partial_t \phi) + (\partial_y c)(\partial_x \partial_t \phi) + (\partial_x c)(\partial_y c)(\partial_t^2 \phi) + (\partial_x \partial_y c)(\partial_t \phi)$$

$$\begin{aligned}(\partial_x \partial_t \phi)(\partial_y \partial_t \phi) \to (\partial_x \partial_t \phi)(\partial_y \partial_t \phi) + (\partial_x c)(\partial_t^2 \phi)(\partial_y \partial_t \phi) \\ + (\partial_y c)(\partial_t^2 \phi)(\partial_x \partial_t \phi) + (\partial_x c)(\partial_y c)(\partial_t^2 \phi)^2\end{aligned} \tag{3.2}$$

As long as the last $\partial_x \partial_y c$ term vanishes, the transformations of the first and second term cancel exactly and the action is invariant.

The equations of motion for this theory are

$$\partial_t^2 \phi \mp g \left[(\partial_t^2 \phi)(\partial_x \partial_y \phi) - (\partial_x \partial_t \phi)(\partial_y \partial_t \phi)\right] + \frac{\partial V}{\partial \phi} = 0. \tag{3.3}$$

The presence of a term linear in $\partial_x \partial_y \phi$ could potentially give rise to instabilities in the theory at it has no definite sign.

We can write down similar theories which only conserve the dipole of energy, or equivalently where the symmetry shifts $t$ by linear functions $c(\vec{x}) = \vec{a} \cdot \vec{x} + t_0$. For example the Lagrangian

$$\mathcal{L} = \frac{1}{2}(\partial_t \phi)^2 + \frac{g}{3}\phi \left[(\partial_t^2 \phi)(\partial_i \partial^i \phi) - (\partial_i \partial_t \phi)(\partial^i \partial_t \phi)\right] - V(\phi). \tag{3.4}$$

has this symmetry. In fact this reduced symmetry is the Carollian limit of the Poincaré group [7–9], and this field theory constitutes a new non-trivial Carollian scalar field theory. The connection between fractonic symmetries and Carollian dynamics have been explored before by [10], and may provide a starting point to understand quantization of these field theories and coupling these theories to a non-flat metric.

Additionally, in the absence of the potential term, the Lagrangian (3.4) is reminiscent of the Lagrangian of scalar Galileon theories [12]. The Galileon Lagrangian, of course, is Lorentz-invariant and invariant under arbitrary shifts, $\phi(x^\mu) \to \phi(x^\mu) + a_\mu x^\mu + b$, whereas ours is explicitly nonrelativistic. Furthermore, our Lagrangian does not have the potential instabilities of (3.1) as $\partial_i \partial^i \phi$ has a definite sign.

## 3.2 Clock Field

The equations of motion following for (2.1) read

$$
\begin{aligned}
\partial_t \left( \dot\phi_1 + i\chi_i \partial_i \phi_2 \right) &= i\partial_i \left( \chi_i \dot\phi_2 \right) - \frac{\partial V}{\partial \phi_1}, \\
\partial_t \left( \dot\phi_2 - i\chi_i \partial_i \phi_1 \right) &= -i\partial_i \left( \chi_i \dot\phi_1 \right) - \frac{\partial V}{\partial \phi_2}, \\
\chi_i &= i \left( \dot\phi_2 \partial_i \phi_1 - \dot\phi_1 \partial_i \phi_2 \right),
\end{aligned}
\tag{3.5}
$$

As long as the potential in (2.1) vanishes, the theory has an additional symmetry: shift invariance of $\phi$ by constants. This symmetry guarantees a solution of, say, the form

$$
\phi_1 = t, \quad \phi_2 = 0.
\tag{3.6}
$$

$\phi_1$ acts as a clock field, that is we can read of time from the value of $\phi_1$. We can expand around this solution by introducing a field

$$
\phi_1 = t + T(t, x, y).
\tag{3.7}
$$

When written in terms of $T$, the action still is invariant under the subsystem spacetime symmetry (1.2), but this time the action on spacetime has to be augmented by a shift of $T$. We still have a locally conserved charge, but this time it is a mixture of an internal charge and the energy density.

## 4 Future Directions

We have written down what appears to be the simplest model of a field theory with subsystem spacetime symmetries. There remain clearly many interesting questions that should be addressed. Among them:

- **Lattice realizations:** Fractons, which often come along with subsystem symmetries, started out as solvable lattice models. Only later was their continuum field theory

understood. For the spacetime subsystem theories we started in the continuum. It would be interesting to understand whether lattice versions of our theory exist, with a time translation symmetry that allows different time translations on different sites. Of course this can be done trivially if the theories on separate sites are decoupled, but our continuum theory suggests that it should also be possible to do this in a theory with non-trivial nearest neighbor interactions. Unfortunately, any attempts to directly discretize our Hamiltonian is hampered by its non-linear form. From the field theory point of view, it is natural to start with a local Lagrangian and let the Hamiltonian be whatever it needs to be. To connect to the lattice, it may be better to start with a simpler Hamiltonian already in the continuum.

- **Higher order corrections:** Clearly, spacetime subsystem symmetries are highly restrictive. It would be interesting to spell out the most general action consistent with these symmetries, including higher order corrections.

- **Quantization:** So far our entire analysis has been classical. It would be interesting to understand quantum theories with spacetime subsystem symmetries.

- **Connection with integrable system:** The Lagrangians we consider have an infinite number of conserved charges, so perhaps they define an integrable system. Unfortunately the Hamiltonian and symplectic structure defined by the Poisson bracket are unwieldy, and we have not been able to find a second symplectic structure to make the system integrable [13]. It would be interesting to understand if these models are in fact intergrable, or if they are somehow related to integrable models.

We hope to address at least some of these points in the future.

## Acknowledgments

The work of SB, AK, AR and HS was supported, in part, by the U.S. Department of Energy under Grant DE-SC0022021 and by a grant from the Simons Foundation (Grant 651440, AK). The work of JD was supported by NSF Grant PHY–2210562.

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
