# Peer review of "Spacetime Subsystem Symmetries"

_SciPost Physics_

## Round 2 · Referee Report · Anonymous · 2023-11-19

Strengths

A simple field theory that is easy to analyze and exhibits novel physics

Weaknesses

1- Sparse on analysis of the field theory including classifaction of simple solutions
2- Suggests directions for more general systems with spacetime subsystem symmetries but does not pursue them in any detail
3- The organization of the sections could be improved, e.g. an analysis on the space of solutions for the model studied in section 2 appears instead in section

Report

The paper presents a very simple model of two interacting scalar fields whose action is invariant under a novel spacetime symmetry of local shifts in the time coordinate. As a consequence of this symmetry, the Hamiltonian density is conserved at every spacetime point and therefore there exists an infinite number conserved quantities. As the authors point out, despite the local conservation, this theory still exhibits dynamics because the momentum density satisfies a dynamical conservation equation.

The spacetime subsystem symmetry that the authors study is a specific choice of diffeomorphism with respect to the time coordinate. It is worth noting that general relativity coupled to matter also exhibits the subsystem symmetry studied by the authors, but GR is more constrained by allowing generic diffeomorphisms. As a consequence, we should expect a relaxed version of the Hamiltonian constraint of general relativity and indeed we see that instead of H=0 on shell, now dH/dt = 0. The Hamiltonian constraint is a consequence of diffeomorphism invariance and not relativity, so the fact that the work studies a non-relativistic limit does not play a large role.

The authors do a good job of discussing the Noether currents associated with the symmetry. However, I think the work could benefit from discussion of the algebra of conserved charges. In particular, because the authors discuss a potential direction for the hydrodynamics of spacetime subsystem symmetries, where a microscopic model may not be available, it would be useful to understand such symmetries from an algebraic perspective. For instance, despite the total momentum being conserved in time, it does not seem to be invariant under the symmetry (1.2).

The second suggestion is to elaborate on the space of solutions. Obviously, without specifying the scalar potential, this is difficult in generality. Nevertheless, the reader would benefit from studying a few simple cases, like $V=0$ and $V = m_1^2\phi_1^2 + m_2^2\phi_2^2$. In both cases, a generic solution is readily obtained. The latter choice of potential is particularly useful because it suggests that in some cases, the sign of the quartic term in the Lagrangian density must be flipped in order to have real solutions. For instance, if

\begin{equation}
\mathcal{L} = \frac{1}{2}\dot{\phi}_1^2+\frac{1}{2}\dot{\phi}_2^2 + \frac{g}{2}(\dot{\phi}_1\partial_i\phi_2 - \dot{\phi}_2\partial_i\phi_1)^2 - m_1^2\phi_1 -m_2^2\phi_2.
\end{equation}

Then, there exists a solution
\begin{align}
\phi_1(t,x) &= i\frac{1}{\sqrt{g}}\frac{m_2}{m_1}\left(a(x)+b t\right), \\
\phi_2(t,x) &= c_1\cosh\left(\frac{\sqrt{2}m_1 x}{b}+\varphi\right) + i \frac{m_1}{m_2}\phi_1(t,x)\sinh\left(\frac{\sqrt{2}m_1 x}{b}+\varphi\right)
\end{align}
which is seen to be real for $g<0$. Conveniently, this has $\mathcal{H}>0$ as $|x|\to\infty$.

For this reason, a more thorough discussion of the space of solutions would be useful. When are solutions real? When are they complex? How is this reflected in the Hamiltonian density and momentum density. Such a discussion would also elucidate why the momentum not invariant with respect to the spacetime symmetry.

Finally, the solutions for the model in section 2 are discussed in section 3.2. This seems out of order and I would suggest moving the section.

Requested changes

1- Discuss algebra of charges
2- Discuss the space of solutions for simple potentials
3- Move 3.2 to section 2

---

## Round 2 · Referee Report · Anonymous · 2023-12-18

Strengths

1- Pedagogical discussion of a simple toy model.

Weaknesses

1- Lacks sufficient investigation of the toy model.
2- Does not discuss differences/similarities with field theories featuring internal spacetime symmetries, besides the similar structure of the symmetry transformation.
3- Connection to Carroll-invariant field theories is not explored in sufficient detail.

Report

The paper proposes simple toy models with "spacetime" subsystem symmetries, extending the recent discussion of "internal" spacetime symmetries appeared in the context of immobile fracton excitations. The paper also draws certain parallels with Carroll-invariant field theories, in particular highlighting how Carroll boosts form the simplest example of spacetime subsystem symmetries. The task undertaken by the authors is a crucial step for furthering our understanding of exotic field theories with immobile or restricted-mobility excitations.

While the model(s) proposed by the authors are certainly novel, I find the paper to be extremely lacking in details and investigation. For example, the authors state the conserved Noether currents for their model, but do not discuss how the conserved currents transform under their newly proposed symmetry or whether the conservation equations are invariant under the said symmetries. This knowledge is quintessential for studying transport properties of these models, as the authors propose to do. I also fail to see how section 3.2 is a generalisation of the model discussed by the authors, as it seems to just be a field redefinition.

Along similar lines, I miss a comparative discussion between "spacetime" and "internal" subsystem symmetries. How much of what we know about internal subsystem symmetries extends or does not extend to spacetime subsystem symmetries? I also find a detailed comparison with Carrollian field theories missing. I gather from author's comments that Carroll-invariant field theories are the best known examples of systems with spacetime internal symmetries. In that sense, it will be interesting to discuss in detail what physical signatures of the new model proposed in the paper transcend beyond what has been observed for Carrollian field theories. For example, the only characterising feature of "spacetime" subsystem symmetries I see in the paper is a locally conserved energy density. However, as I understand, this is also observed in Carroll-invariant field theories, so it would be interesting to see what added constraints do the stronger requirement of generic "spacetime" subsystem symmetries imply.

Finally, authors discuss subsystem symmetries that act on the time-coordinate "t". One might also imagine a similar construction for space coordinates, to make these true "spacetime" symmetries. Some comments along these lines will be appreciated.

In summary, while the subject matter of the paper is certainly interesting, I find the paper lacking key analysis and discussion to warrant publication in its present form.

Requested changes

1- Add more analysis of the simple model proposed by the authors. Discuss the physical implications of spacetime subsystem symmetries on observables such as conserved Noether currents.
2- Discuss the similarities and differences with internal subsystem symmetries.
3- Add more comments on comparison with Carrolian-invariant field theories and how the models discussed in the paper have more general physical properties.

---

## Editorial Decision

awaiting_resubmission